# BAYES ALWAYS WINS THE LOTTERY IN MONTE CARLO

## ABSTRACT

Most contemporary neural networks suffer from huge model sizes requiring prohibitive storage and computational resources for training, making their use difficult on edge devices. Neural network pruning aims to address this problem. The lottery ticket hypothesis is a sparse pruning method that can reduce the network size significantly with minimal accuracy loss. However, the original initialization sample is needed in order for the "winning ticket" to train to the same accuracy after pruning. We present a novel approach utilizing Hamiltonian Monte Carlo to always win the lottery by training Bayesian neural networks with lottery-ticket generated pruning masks from any initialization. Our first key finding is to establish a generalized framework for training lottery ticket pruned networks, independent of specific initialization samples, with a Bayesian-based theoretical grounding containing convergence guarantees that ensure the optimal initialization distribution is found. Second is that networks trained using this framework achieve predictive performance equivalent to or exceeding that of networks initialized with the lottery ticket initialization sample. Finally, we investigate whether stochastic gradient based Bayesian methods can achieve similar performance as Hamiltonian Monte Carlo. Result highlights include that on LENET300-100 networks on CIFAR-10 using Hamiltonian Monte Carlo, we observed a best-case accuracy improvement of 5% over random initialization samples and 3% over the original lottery-ticket initialization sample, highlighting the capabilities of Bayesian methods for training pruned networks.

## 1 INTRODUCTION

Current neural networks suffer from an explosion of parameter size, with variants such as VGG-16, ResNet, and modern transformer-based LLM models taking hundreds of millions, sometimes billions, of parameters to train Simonyan & Zisserman (2015); He et al. (2016); Brown et al. (2020). Pruning methods have proven a reliable method for reducing the parameter amount, while retaining performance Cheng et al. (2024). Some pruning methods can even take advantage of certain properties of neural networks to obtain significant performance. One such method is the Lottery Ticket Hypothesis Frankle & Carbin (2019).

Neural networks are highly reliant on the initialization of the parameters, and certain initializations may be more advantageous than others, starting in a better loss neighborhood than others. The Lottery Ticket Hypothesis takes advantage of this to prune a neural network. The hypothesis states: **There exists a subnetwork within any sufficiently large neural network which, when trained from a specific initialization, achieves the same performance as the full network.** Importantly, the initialization affects the resulting subnetwork, and the pruning process can be different for different initializations.

This can make evaluation of what makes the generated pruning masks work so well difficult. The current theory is that these masks perform well with the given initializations because these initializations start nearby a very strong local, or potentially the global minimum Frankle & Carbin (2019); Evci et al. (2022).

Precisely this means that when we are looking to minimize the loss of a neural network (eq. 1),

$$\min L(y, \hat{y}), \tag{1}$$

where our output will be an n-layered neural network, with a given layer $i$ being the weighted output of a previous layer and an activation function $\sigma$, as given by eq. 2,

$$X_i = \sigma(A_i X_{i-1} + b_i). \tag{2}$$

In order to find the minimum, the weights of each layer ($A_i$) are trained by gradient descent, which updates each weight $a_{i,j}$ in the weight matrix $A_i$ by eq. 3,

$$a_{i,j}^k = a_{i,j}^{k-1} - \eta \nabla X_{n_{i,j}}(a_{i,j}) \tag{3}$$

where $\eta$ is the learning rate (can be adjustable), and the gradient is found using backpropagation. Since this is gradient descent, we will need to initialize a "guess" for $a_{ij}^0$. This guess is usually initialized by drawing from a probability distribution, typically some form of uniform distribution, such as eq. 4,

$$A_i^0 \sim U(-c, c) \tag{4}$$

Where $c$ is some constant. Assuming we have two different draws from this distribution, such as $A_i^0 = p_0 \sim U(-c, c)$ and $A_i^{0'} = p_1 \sim U(-c, c)$, then if $p_0$ is used as our initialization and a pruning mask generated from it using iterative magnitude pruning Frankle & Carbin (2019), the performance of training using that mask would be superior if using $p_0$ rather than using $p_1$. This also applies to any arbitrary draw such that $p_i \neq p_0$.

However, this performance difference only arises because of gradient descent, since eq. 1 is non-convex if using a neural network. Since gradient descent is so linked to which initialization is used, it would be beneficial to take a more general approach that is not linked to a specific initialization.

Instead, we can look at the entire initialization space at once. Since an initialization is drawn from a probability distribution (in the example, a uniform one), it is natural to treat this in a Bayesian manner. We can then use Bayesian techniques to find the true posterior, which will be the probability distributions which, when drawn from, will give us the most optimal choice of weights, and thus the highest probability of minimizing the loss.

In order to achieve this, we will utilize Bayesian neural networks trained using Hamiltonian Monte Carlo (HMC) Neal (1996). HMC is a Markov chain Monte Carlo method and has proven guarantees of convergence to the true posterior distribution, which will allow for achieving the same performance on a pruning mask generated by $p_0$, without needing $p_0$. This would then allow for the analysis of randomly-generated pruning masks, as the specific unknown initialization needed to obtain full performance is no longer needed. We will also compare with Stochastic Variational Inference (SVI) Hoffman et al. (2013), a non-sampling based Bayesian technique, which while having fewer guarantees than HMC, is faster and may be more practical in real world situations.

## 2 RELATED WORK

This work will use the lottery ticket hypothesis of Frankle & Carbin (2019). We look to further add evidence for the performance of lottery ticket networks. Approaches to study why lottery tickets work well include Evci et al. (2022); Zhang et al. (2021b), which investigates the training dynamics of lottery ticket networks through techniques such as gradient flow, sample complexity and the optimization landscape. There are also works that investigate the pruning masks themselves, aiming to show important aspects of the mask structure Zhou et al. (2019); Su et al. (2020).

A similar work to ours has been done with evolutionary strategies, showing that they can be used to train lottery ticket networks to similar performance Lange & Sprekeler (2023). We hope to improve upon this result, by demonstrating that we can achieve similar performance using a Bayesian technique with theoretical optimality conditions, that is, it is sure to converge to a true posterior given

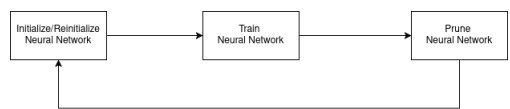

Figure 1: How to find a lottery ticket network.

enough samples, as well as showing that Bayesian neural networks can be viewed as a generalized initialization for deterministic neural networks.

Furthermore, there are works investigating a "strong lottery ticket hypothesis", which states that pruning a network at initialization can be equivalent to training that network Malach et al. (2020); da Cunha et al. (2022). However, these works do not consider training the networks and as such do not consider what about the initialization of lottery ticket networks makes them special. In this work, we want to see if we can replicate the success of the original lottery ticket networks with only the pruning mask and training. If it is possible to replicate lottery tickets with only the pruning masks and training, then this opens up avenues towards investigating the pruning masks themselves and can be a stepping stone towards answering the question of how important the structure of the masks is.

Finally, much work has gone into finding tickets (that is, pruning masks) more efficiently, using methods such as continuous sparsification Savarese et al. (2020), utilizing the strong lottery ticket hypothesis to find good masks without training networks Wang et al. (2020); Altmann et al. (2024), or even using a subset of the data to decrease computational load Zhang et al. (2021c).

## 3 METHODOLOGY

First, we will briefly reintroduce the process of making a pruning mask using the lottery ticket hypothesis. We will use the process as defined in the original work of Frankle & Carbin (2019). This process consists of first initializing a neural network, saving that initialization, and training it. Assuming $n$ total pruning iterations with a target pruning percentage $p$, the network is then pruned $p^{\frac{1}{n}}\%$ using magnitude pruning. The weights are then reset to the saved initialization, and the process is repeated until the desired pruning target is met. This process can be seen in Figure 1.

Once obtained, a mask will be able to retain performance, but only with the specific initialization used to generate the mask. Random initializations will not be able to meet the same performance. Frankle & Carbin (2019) speculated that, for the mask generating initialization, that mask is fit to prune weights that are not near strong optima, while those weights that are nearby are kept.

### 3.1 HAMILTONIAN MONTE CARLO

When using MCMC based methods, it would be natural to first try standard random-walk MCMC. While this has the best ability to explore the state space, it is also extremely slow, and moreover, the way it is implemented cannot take advantage of the fact that the parameters to be optimized are part of a neural network, and so there are certain ways in which this information can be exploited. With neural network parameters, we can get gradient information since it is a continuous, rather than discrete, optimization setting. Hamiltonian Monte Carlo (HMC) can take advantage of this gradient information to help guide the search Neal (1996). A brief description of the algorithm is as follows: Suppose we have a probability distribution $f(x)$ we wish to model. We can model it via MCMC by taking samples, and accepting or rejecting those samples. In HMC, we can model the dynamics using a Hamiltonian, which will propose samples in an energy conserving manner, ensuring a higher acceptance of new samples and a more efficient exploration of the posterior. This Hamiltonian is defined by equation 5.

$$H(x,p) = U(x) + \frac{1}{2}p^T M^{-1}p, \tag{5}$$

where $U(x)$ is the potential energy of the system at state $x$, which is $-\ln f(x)$, $p$ a momentum variable (with $p(0)$ drawn from a standard normal distribution) and $M$ a mass matrix (the mass matrix

represents how well the velocity and the kinetic energy are related. While it is best to analytically solve this, it is difficult, if not impossible for neural networks, and so the software used finds an approximation during a warmup period).

In any MCMC algorithm, we need to propose a new state to move to (in the case of a neural network, a new set of parameters), and move to it if the state is better, or probabilistically move there if it is worse. In each step, the standard MCMC proposes a nearby state, where the distance to the state is determined by a proposal distribution, typically Gaussian. HMC on the other hand, will take $L$ steps of size $\Delta t$. The current state $x$ and the momentum $p$ are then updated at each step $\Delta t$ by equation 6:

$$p(t + \frac{\Delta t}{2}) = p(t) - \frac{\Delta t}{2}\nabla U(x(t))$$
$$x(t + \Delta t) = x(t) + \Delta t M^{-1} p(t + \frac{\Delta t}{2}) \tag{6}$$
$$p(t + \Delta t) = p(t + \frac{\Delta t}{2}) - \frac{\Delta t}{2}\nabla U(x(t + \Delta t)),$$

which we can run for $L$ times to get the next potential state $x'$, and the momentum thereof $p'$. We will then draw a random number uniformly between 0 and 1, and if that uniform number is lower than our acceptance criteria, given by equation 7, we will accept the new state. HMC is reversible and, as such, it will converge to the true posterior distribution Neal et al. (2011).

$$\alpha(x, x') = min\left\{1, \frac{e^{-H(x',p')}}{e^{-H(x,p)}}\right\} \tag{7}$$

Additionally, since $L$ is a hard-to-determine hyperparameter, we used the No U-turn sampler (NUTS) to automatically determine a good $L$ Hoffman et al. (2014). NUTS obtains a good choice of hyperparameter $L$ by continuously taking steps with two samplers, one forwards and one backwards using the HMC update rules, until the U-turn criterion is met. The new next state is then sampled from the path.

## 3.2 HMC FOR TRAINING BAYESIAN NEURAL NETWORKS

A Bayesian neural network will treat the parameters of the network as distributions instead of singular numbers. If we look again at eq. 4, we can decompose this into each individual weight as a sample from a distribution (eq. 8)

$$a_{i,j}^0 \sim U(-c, c). \tag{8}$$

We can take this a step further and instead assign each $a_{i,j}^0$ to a random variable, and instead of taking a sample from the distribution, the weight itself is a distribution, and we can train this distribution. We can treat the right side of eq. 8 as the prior distribution, and use this entire distribution as our starting point to find the true posterior for each $a_{i,j}^0$.

Once the posterior is found, or a good approximation thereof, we can then use these as the initialization distribution, and not do any backpropagation based gradient descent training. By taking enough samples, the average of these should give us the most likely weight values that lead to the best performance. Because HMC is an MCMC based technique with convergence guarantees, then it should, again given enough time, find these values regardless of our initial prior. It should be noted though that the prior will have an effect on how long this takes, with some leading to faster convergence than others, but unlike with gradient descent where a sample has an outsized effect on convergence to a global optima, HMC will converge to the true posterior, and therefore will give optimal weights for the network regardless of the initial initialization distribution used.

## 3.3 STOCHASTIC VARIATIONAL INFERENCE

An alternative to MCMC based methods is variational inference, an often faster albeit approximative method to perform inference over large datasets and models.

Variational Inference operates by choosing a family of tractable parametrized distributions $q(w)$ and modifying its parameters to best approximate the true posterior $p(w \mid D)$. The measure of closeness of the approximation is given by the Kullback-Leibler (KL) divergence, where minimizing the divergence corresponds to maximizing the Evidence Lower Bound (ELBO) (eq. 9), and consequently approximating the true posterior and thus acting as the objective function. Stochastic Variational Inference (SVI) Hoffman et al. (2013) specifically scales this operation by optimizing the ELBO using mini-batches of data, but does not have the same general convergence guarantees that Markov Chain Monte Carlo methods carry.

$$\log p(D) = \underbrace{\mathcal{L}(q)}_{\text{ELBO}} + \underbrace{\text{KL}\big(q(w) \,\|\, p(w \mid D)\big)}_{\geq 0} \tag{9}$$

Bayes By Backprop Blundell et al. (2015) is an efficient practical implementation of SVI on Bayesian neural networks. It works by propagating a single sample from the weights forward through the network, and uses that sample to compute gradients of the ELBO, then updates the parameters of the probability densities over the weights through backpropagation. The objective function is the negative ELBO seen in eq. 10, where the first term is the likelihood term and the second is the KL divergence.

$$\mathcal{L}(q) = E_{q(w)}\left[-\log p(D \mid w)\right] + \text{KL}[q(w) \,\|\, p(w)] \tag{10}$$

For classification tasks in particular, the likelihood term becomes the cross entropy (eq.11), which is the cross entropy loss. The loss function then assumes the form in eq. 12.

$$E_{q(w)}\left[-\log p(D \mid w)\right] = E_{q(w)}\left[-\log p(y \mid x, w)\right] \tag{11}$$

$$min(\mathcal{L}) = CE + KL \tag{12}$$

## 4 Experiments

To test whether we can match the accuracy of the lottery tickets using HMC methods, we set up a variety of tests with several different types of neural network. We tested on both small and larger networks. The networks we tested consisted of a LeNet300-100 model (fully connected two hidden layers with 300 and 100 neurons), and LeNet5 (a convolutional network) LeCun et al. (2002). For HMC, larger networks were out of reach at this time, which we discuss more thoroughly in the limitations. However for SVI we did additionally test Resnet-18, containing 11 million parameters He et al. (2016).

LeNet300-100 and LeNet5 were tested on MNIST LeCun et al. (2010) and CIFAR-10 Krizhevsky (2009). For prior distributions, we used several different priors, such as a standard normal distribution and uniform distribution with bounds of -1 and 1. Additionally, since it is suspected that lotto ticket networks are already nearby the optimum, we tried priors that mimic the initialization strategies used to generate initializations, namely a glorot normal and glorot uniform distribution, which we modified to have a slightly wider variance Glorot & Bengio (2010).

The networks trained using HMC all had 200 warmup samples (samples that are used to find an appropriate mass matrix $M$, etc), and then either 1400, 1800, or 2800 samples were taken, depending on the network and the dataset.

For the networks using SVI, each fully connected layer was assigned a normalization factor for the KL term in eq. 12. Without such a normalization, the much larger KL term would end up absorbing any gradient contribution given by the CE term. To find the optimal constants for the normalization, we employed a population-based training method where 50 randomly initialized models were compared and the best resulting models hyperparameters were then tested. As a control, we also tested bayesian neural networks with SVI with the default hyperparameters.

The pruning masks used were generated by the lottery ticket hypothesis using the iterative magnitude pruning procedure as shown in figure 1.

All methods were trained using these masks. The SVI based Bayesian networks and comparison deterministic networks used 20 different random initializations. Additionally, a comparison with the lottery ticket initialization was used for deterministic networks. Each comparison Lenet300-100 and Lenet-5 network was trained for 100 epochs for CIFAR-10 and 50 epochs for MNIST, with further training yielding no benefit. Due to resource limitations, Resnet-18 was limited to 200 epochs.

## 5 RESULTS AND DISCUSSION

For Lenet300-100, HMC training consistently outperformed random initializations, and met or exceeded the performance of the initializations used to generate the masks (lottery initialization). We can see in figure 2 that for the MNIST data, the average random initialization can be anywhere from $0.5\%$ below the lottery ticket, and can be as much as $7\%$ below. We also see HMC can achieve anywhere from the same performance to $0.5\%$ boost as compared to the lottery ticket initialization. We see the largest performance boost on LeNet300-100 on CIFAR-10, where the HMC method outperforms the lottery ticket initialization by up to $4\%$, which is a larger performance difference than between the lottery ticket initialization and random initializations, as seen in figure 3.

We also display a table of results for the largest pruning amounts in table 1, where we can clearly see that HMC training is outperforming both lottery ticket initializations and random initializations, with the one exception of the average cross-entropy loss being slightly lower for LeNet5. On this note, we also want to mention that convolutional networks like LeNet5 had some variance in their final performances over multiple runs, where even after 3000 samples for a run (200 warmup + 2800 samples), where between runs the accuracy could dip by as much as $5\%$. This suggests that the Markov chain had not quite converged for LeNet5, and even more samples would need to be taken in order to get to the true posterior.

Running HMC is very expensive, so we could not realistically take more samples. Moreover the point of using HMC is because it has convergence guarantees, not necessarily that it is the fastest or most efficient Bayesian technique. To that end, we did try substituting HMC with SVI, since while it does not have the convergence guarantees that HMC has, SVI is much faster as it is not sampling based. This training was faster, completing in an average of 2 hours rather than 2 days, after the best performing hyperparameters were identified. After 50 epochs performance would be consistently slightly below the lottery ticket performance for LeNet300-100 on MNIST on the 99% pruning mask, with 96.74% accuracy against 97.05% with lottery ticket initialization (and an average of 96.85% with random initializations). The same network performed performed comparatively better on CIFAR, with an average of 53.27% against 53.3% with lottery ticket initialization. In both cases the hyperparameter-optimized networks (whereby only proportional weight of KL-divergence vs. CE loss) performed far better than SVI with default hyperparameters (96.5% on MNIST and 52.4% on CIFAR). This is a good indication that other Bayesian techniques may still work, even if they do not have the guarantees that HMC has. We additionally see that SVI can scale as well, owing to the stochastic-gradient based nature, with SVI being able to train a Resnet-18 on Cifar-10 in a similar amount of time to deterministic methods, while even exceeding the lottery ticket performance (HMC could not even load Resnet-18 due to GPU memory limitations).

Although SVI does not offer the convergence guarantee given by HMC, it experimentally achieves comparable performance for small-sized networks, and can achieve better results for larger networks while maintaining similar computational time to that of deterministic methods. A possible explanation for its performance could be that SVI averages a collection of sampled weights to generate an expectation that then is optimized. This optimization is effectively a smoothed version of the loss landscape, which is easier to optimize and less sensitive to local points of optimality that define the theory behind LTH.

## 6 LIMITATIONS

### 6.1 DATASET AND MODEL SIZE

Our experiments were limited to relatively small (in modern use-cases) datasets, specifically CIFAR-10 and MNIST. Additionally, the models that were used were relatively small, with the largest model being just under half a million parameters. This is primarily because the use of HMC with the NUTS

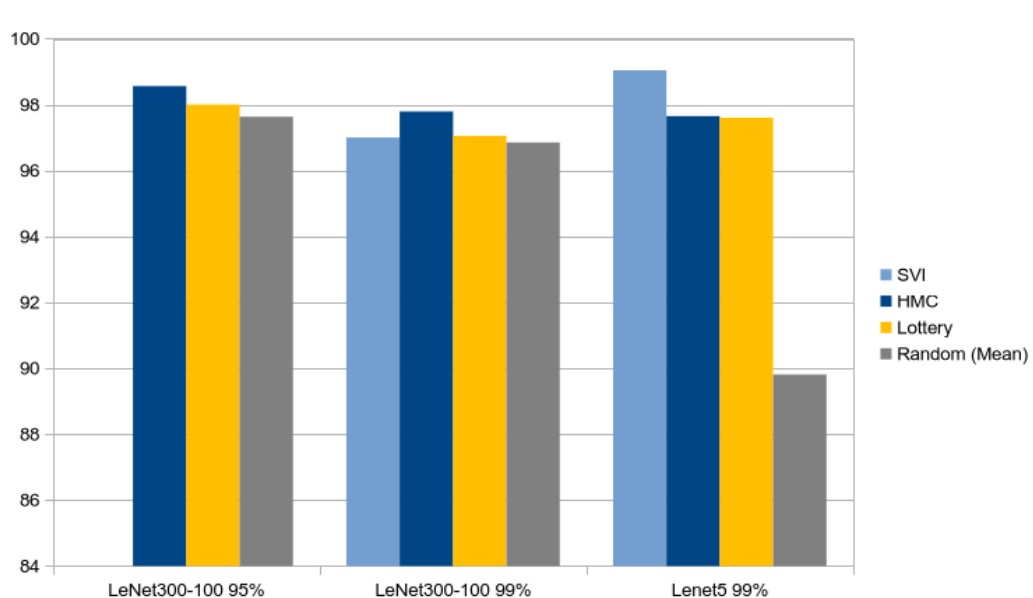

Figure 2: Accuracies on MNIST of models trained with HMC, lottery initialization, random initializations, and SVI. X% denotes how much the model was pruned, with 99% being a model with 1% parameters left.

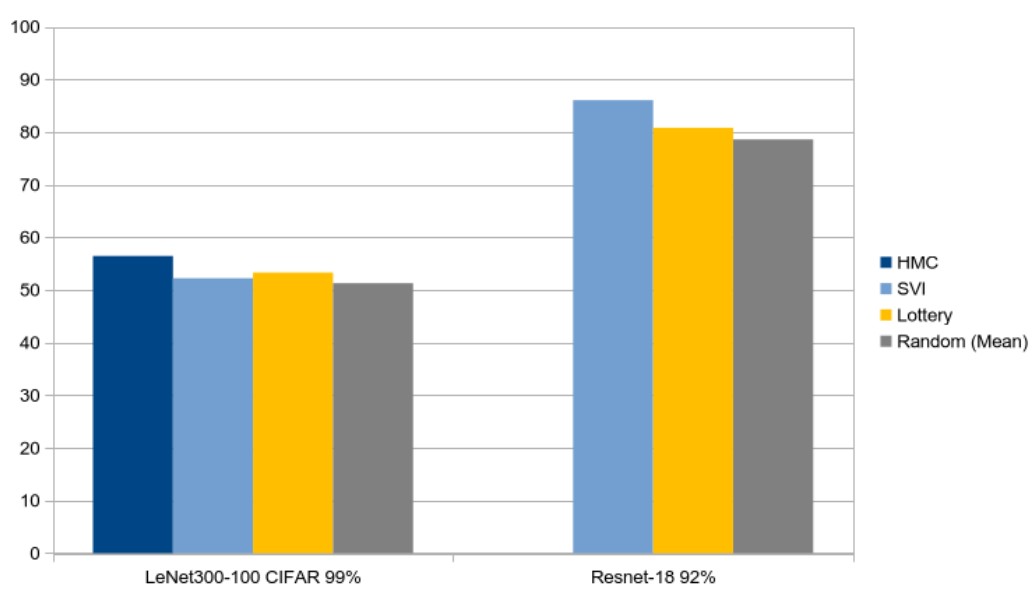

Figure 3: Accuracies on CIFAR of models trained with HMC, lottery initialization, random initializations, and SVI. X% denotes how much the model was pruned, with 99% being a model with 1% parameters left.

| | Lenet300-100 M | Lenet300-100 C | Lenet5 M | Resnet18 C |
|---|---|---|---|---|
| HMC Acc. | **97.79%** | **56.45%** | 97.65% | NA |
| Lotto Acc. | 97.05% | 53.3% | 97.6% | 80.79% |
| SVI Acc. | 97% | 52.22% | **99.04%** | **86.04%** |
| Rand. Acc. | $96.85 \pm 0.79\%$ | $51.28 \pm 0.38\%$ | $89.81 \pm 1.39\%$ | $78.59 \pm .08\%$ |
| HMC Loss | **0.0981** | **1.2519** | 0.0747 | NA |
| Lotto Loss | 0.1028 | 1.4559 | 0.0689 | 4.0451 |
| SVI Loss | 0.1162 | 1.4092 | **0.0618** | **1.2957** |
| Rand. loss | $0.117 \pm 0.01$ | $1.376 \pm 0.01$ | $0.333 \pm 0.05$ | $4.062 \pm 0.01$ |

Table 1: Accuracies/Loss on all the different models at the highest tested pruning percentage (92% pruned for Resnet18, and 99% pruned for the rest). M is MNIST dataset, C is CIFAR-10.

sampler requires the full dataset to be processed at once, which if combined with larger models, requires significant memory to properly utilize, with tests of CIFAR-10 combined with a 1.1 million parameter convolutional network quickly running out of memory on a 48GB GPU. Techniques such as Stochastic-Gradient HMC (SG-HMC) Chen et al. (2014) could alleviate this by enabling mini-batching of the data. While not an MCMC based method, we did test the performance of SVI, and still saw performance that beat random initializations, while in some cases meeting or exceeding the performance of the lottery initialization, so long as some hyperparameter optimization of the normalization terms was performed. To test the scalability of SVI, we also tested it's performance on Resnet-18. While SVI does not have the same theoretical guarantees as HMC, the performance of it suggests stochastic techniques could be a promising direction towards alleviating this pressure. However, stochastic based methods do not guarantee the convergence of the posterior, the convergence being a property we desire Betancourt (2015).

## 6.2 Training Time

Training the networks is also significantly more costly than standard gradient-descent techniques on deterministic neural networks. Whereas training the deterministic versions could take anywhere from 5 to 30 minutes, training using HMC could take anywhere from several hours to several days, depending on the dataset and model used. This is mainly due to HMC needing multiple gradient calculations to take a sample, as it models the Hamiltonian dynamics. Additionally, there is a Metropolis-Hastings step to accept or reject the new state, so not every sample is necessarily used either, making sampling take even longer to get to the posterior. However for SVI, the training time was equivalent to the deterministic settings, while often meeting, and sometimes even exceeding the performance of the lottery ticket initialization.

## 7 Future Work

The primary motivation was to show that it is possible to get lottery-ticket performance from lottery-ticket pruning masks without needing the initialization used to generate the mask. We were surprised to see just how well deterministic pruning masks could perform when applied to Bayesian neural networks, and hope that this work will enable better exploration of what makes pruning work. We hope to see more about how important the structure of the mask, or what is pruned, matters. Especially with the lottery ticket hypothesis, while previously if a random mask had poor performance, it would be inconclusive to show that it does not work while only using gradient descent, but by using HMC, if the mask still fails to perform, it should give better evidence to show that the structure of the mask is lacking. There has been some work in this direction Su et al. (2020), and we hope that this work can help further study these effects.

A secondary potential future application is to see whether stochastic-gradient MCMC methods could work. HMC is well studied, and the convergence properties are known, hence our choice to use it, but we make no claim that this is the best MCMC algorithm to use. There have been several recent advances in SG-MCMC, such as hybrid SG-HMC Zhang et al. (2021a), and also a class of several alternative methods using stochastic gradient Langevin dynamics Welling & Teh (2011). Through SVI we saw stochastic-gradient methods can achieve considerable performance, we reason these

could be useful avenues to explore, as stochastic-gradient methods could enable the exploration of larger datasets, and larger networks.

Finally, with the surprisingly good performance of deterministic pruning (pruning masks created by and for non-Bayesian neural networks) on Bayesian neural networks, it may be prudent to look at how other deterministic pruning techniques perform on Bayesian neural networks. Additionally, since Bayesian neural networks do allow for more information in the form of posteriors over the weights, utilizing this extra information to create new pruning techniques could be a valuable path forwards.

# 8 CONCLUSION

We have demonstrated that by using Hamiltonian Monte Carlo, it is possible to break the lottery. We have shown that Hamiltonian Monte Carlo is a viable technique for training networks with a lottery ticket pruning mask, and reaching the same level of performance as training with the initial initialization, showing that this initialization is not necessary. Our method comes with the convergence guarantees of HMC, meaning it is sure to converge to the optimal distribution given enough samples. We demonstrated this on pruning masks of various sparsities on several different networks, showing that HMC can achieve in good cases up to a $4\%$ improvement as compared to the lottery initialization, and as much as $10\%$ compared to random initializations.

We also showed that other Bayesian methods could be a viable way forwards. We showed that SVI methods can lead to results comparable to the lottery ticket initialization, even exceeding them as was the case for Resnet-18, though with the caveat of hyperparameter tuning being necessary for Bayesian networks.

This surprisingly good performance on Bayesian neural networks, even though the pruning masks are from a deterministic source, is because we can treat the problem in a Bayesian manner, and avoid the initialization sample dependency of backpropagation-based gradient descent training, allowing for a more general treatment of the training process, and thus the ability to train on extremely sparse lottery-ticket pruning masks with only the pruning mask itself.

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
