# OpenReview forum: "Bayes Always Wins the Lottery in Monte Carlo"
_ICLR.cc/2026/Conference — ICLR 2026 Conference Withdrawn Submission_

### Official Review · Reviewer_5Ycz · 2025-10-23

**Soundness:** 2
**Presentation:** 1
**Contribution:** 1
**Rating:** 0
**Confidence:** 4

**Summary:**

The presentation of the paper is not good.
In my understanding, the submission studies the following setting:
- You train a neural from a weight initialisation $w\_{\text{init}}$.
- You prune the resulting network after training.
- Then you retrain the network from either a random weight initialisation or from $w\_{\text{init}}$.

Typically, the performance is better when you start from $w\_{\text{init}}$.
However, when you train the network in a Bayesian setting using Hamiltonian Monte Carlo, you get good results also from random weight initializations (different from $w\_{\text{init}}$).

**Strengths:**

n/a

**Weaknesses:**

The presentation of the paper is not good, to a degree that many parts are obscure.
This is not a language issue, it is about a lack of structure and mathematical rigour in the presentation, see below.

The basic research question is not well motivated.
In my understanding, the submission studies the following setting:
- You train a neural from a weight initialisation $w\_{\text{init}}$.
- You prune the resulting network after training.
- Then you retrain the network from either a random weight initialisation or from $w\_{\text{init}}$.

Except giving repeated references the „lottery hypothesis“, the manuscript fails to motivate why this is an interesting setting at all.
If I do pruning, why should I retrain from a random initialisation or $w\_{\text{init}}$ instead of continuing from the weight configuration after pruning (i.e., the learnt weights from before the pruning), that is, perform standard „fine-tuning“ ?


## Experiments:

Most results in Table 1 come from single trials. Thus, there is a risk that the differences are random artefacts.

It is not clear how much the results depend on hyperparameter choices, where I am particular thinking of hyperparameters of the learning algorithm. Do the „Rand. Acc“ vary depending on the learning rate? (Note: when initialising a pruned network using ReLU activation functions, one wants to avoid dead neurons, i.e. I would not initialise the bias parameters with mean zero but to be strictly positive).

As discussed by the authors in section 6.2., HMG can take several days. In the same time period, many(!) standard training trials can be performed - which can also be viewed as some sort of sampling. The standard training should get the same  computational resources as the HMG training, and then (using training or validation loss) the best solution of the many trails should be picked. Another words, HMG should be compared to standard training with random restarts (perhaps with varying learning rates).


## Related work:

The paper considers pruning, and it considers Bayesian neural network training. However, the extensive literature on Bayesian pruning is not discussed in the paper. For example, see

Dustin Wright, Christian Igel, and Raghavendra Selvan. BMRS: Bayesian Model Reduction for Structured Pruning. Advances in Neural Processing Systems (NeurIPS), 2024

and references therein. Although t is not the main topic of the study, it would be interesting to see how Bayesian model reduction techniques such as BMRS perform in comparison.

## Presentation:

The presentation of the background and used algorithms is incomplete, hardly comprehensible, and at least partly wrong.
Te authors fail to explain gradient-based optimization of neural networks („backdrop“) properly.
Clearly the gradient update in eq. (3) is wrong as it is independent of the loss.

Another example:  The paper states „We can then use Bayesian techniques to find the true posterior, which will be the probability
distributions which, when drawn from, will give us the most optimal choice of weights, and thus the highest probability of minimizing the loss.“, which is clearly wrong. If the posterior has not converged/collapsed, then every draw will give a different weight configuration and not each of them can be the „most optimal“. (BTW, are there „optimal“ configurations which are not „most optimal“? .. sorry, couldn’t resist.)

The explanation of Hamiltonian Monte Carlo (HMC) is irritating to read. The manuscript sets out to describe the basic concept of the algorithm. But for which type of readers should phrased such as „which will propose samples in an energy conserving manner“ or „represents how well the velocity and the kinetic energy are related.“ I assume that for most people who do not know HMC, even  if they are experts in MCMC, these phrases explain nothing. Better no description than the one provided.

Equation (7) is not connected to the text, $\alpha$ is not introduced.

**Questions:**

please see my comment in "Weaknesses" above.

---

### Official Review · Reviewer_ngZs · 2025-10-26

**Soundness:** 1
**Presentation:** 1
**Contribution:** 1
**Rating:** 0
**Confidence:** 4

**Summary:**

This paper explores an intersection between the Lottery Ticket Hypothesis (LTH) and Bayesian learning. The authors propose using Hamiltonian Monte Carlo (HMC) to train Bayesian neural networks (BNNs) on pruning masks derived from the LTH, arguing that this approach removes dependence on the original initialization. They also compare HMC with Stochastic Variational Inference (SVI) as a faster alternative.
Experiments are conducted on LeNet300-100, LeNet5, ResNet-18 using MNIST and CIFAR-10, to support the claim that Bayesian training may match or slightly exceed the accuracy of deterministic lottery-ticket baselines.

**Strengths:**

The work connects two well-known areas, LTH and Bayesian inference.

**Weaknesses:**

The paper presents an interesting idea, but the central claims are not convincingly demonstrated.
The theoretical discussion remains at a high level and does not provide sufficient formal grounding for the claimed generality. The experiments are limited to relatively small networks and do not support the broad conclusion that initialization becomes irrelevant. Crucially, the **computational comparison is unbalanced**: HMC training is substantially more expensive than SGD, and this difference is not clearly reported or accounted for.  The paper would be stronger if it offered either new theoretical insights or a more systematic empirical study demonstrating when and why the proposed combination is beneficial.
Moreover, the presentation would benefit from significant revision. The structure can be confusing, with the methodology and background intermixed. Figures and tables could be improved: some are low resolution or missing elements, and do not look professional. There are  stylistic and formatting inconsistencies, such as unstandardized citations and inconsistent capitalization.

**Questions:**

- Please report the computational budget (time or, even better, FLOPS) used for each training method, so comparisons can be fairly interpreted. How does your method perform if one allocates comparable resources to repeating the classical methods and keeping the best result?
- Could you clarify the practical motivation for viewing the lottery ticket as a Bayesian network? What specific new insights into the Lottery Ticket Hypothesis are gained from this Bayesian formulation? Are the resulting tickets transferable?
- How do you envision this method scaling to realistic architectures beyond LeNet and small ResNets?

---

### Official Review · Reviewer_eQhw · 2025-10-31

**Soundness:** 2
**Presentation:** 2
**Contribution:** 2
**Rating:** 2
**Confidence:** 4

**Summary:**

The paper proposes training lottery-ticket-pruned networks (masks) with Bayesian inference (Hamiltonian Monte Carlo (HMC) and Stochastic Variational Inference (SVI)) to remove dependence on the original winning initialization, reporting gains on LeNet-300-100/LeNet-5 and competitive SVI scaling to ResNet-18 on CIFAR-10. The authors position this as a more general alternative to deterministic training that "always wins" via posterior convergence.

**Strengths:**

1. Clear motivation to decouple mask quality from the original initialization using Bayesian training; this is interesting for analyzing mask utility.
2. Practical insight that tuned SVI can be competitive and scale to larger models, offering a usable alternative where HMC is impractical.

**Weaknesses:**

1. The theory in the paper is limited to generic HMC convergence; there is no problem-specific analysis for masked neural posteriors, no mixing/convergence diagnostics, and no guarantees linking posterior quality to recovering or exceeding lottery-initialized performance, which matters because HMC is costly and requires many samples to mix well in practice, as the experiments in the paper show.
2. (Overclaiming) The phrases "always wins" and "sure to converge" are not backed by explicit theory and conflict with the paper's own limitations; without supporting analysis, claims should be narrowed to the tested regimes.
3. The main table should include means ± standard deviations and seed counts for all methods, given observed variability. Confidence intervals and calibration metrics would strengthen conclusions.
4. (Focus drift) The paper initially leans on HMC's convergence rationale but pivots to SVI once HMC hits practical limits, without a deeper discussion of why SVI works here and when it should be preferred.
5. A consolidated hyperparameter table (optimizers, priors, SVI KL weights/normalizations, HMC step size, mass matrix adaptation, warmup, samples) is needed for reproducibility.
6. The paper claims to avoid initialization dependency, but HMC's finite-time efficiency can still depend on starting state and tuning; clarify scope (asymptotic vs. practical) and report warmup, acceptance, and mixing statistics.
7. (Comparative positioning) The stated improvement over evolutionary/gradient-free training is not demonstrated, as there is no head-to-head comparison with the mentioned baseline (Lange et al.), nor any theoretical justification specific to this setting.
8. An end-to-end diagram of the Bayesian masked training pipeline (masking, priors, HMC/SVI steps, prediction aggregation) would clarify the workflow and evaluation protocol.
9. Formatting issues: Usage of "lotto" instead of "lottery" ticket at some places; references are missing parentheses (likely incorrect \cite macro); these are mostly minor issues.

**Questions:**

1. What convergence diagnostics were monitored for HMC (e.g., acceptance rates, step size/mass matrix adaptation), and how do these relate to the ~5% cross-run variance observed on LeNet-5?
2. How many posterior samples were used at test time, how were predictions aggregated (predictive mean vs. logit integration), and how sensitive are the results to warmup length and number of samples?
3. Can the authors provide a head-to-head comparison with evolutionary/gradient-free training (e.g., the cited baseline of Lange et al.), and clarify where Bayesian training improves or differs in practice?
4. Please add means +/- standard deviations (and number of seeds) for all methods, and include calibration/uncertainty metrics to leverage the Bayesian framing.
5. Can the authors include a consolidated hyperparameter table and an illustrative end-to-end diagram of the Bayesian masked training pipeline to improve reproducibility and clarity?
6. Given that HMC's finite-time performance can still depend on initialization/tuning, can the authors clarify the precise sense in which initialization dependence is "avoided", and report warmup/acceptance/mixing statistics accordingly?

---

### Official Review · Reviewer_BQZG · 2025-11-01

**Soundness:** 2
**Presentation:** 2
**Contribution:** 2
**Rating:** 2
**Confidence:** 2

**Summary:**

The paper tries to solve the initialization dependence of the Lottery Ticket Hypothesis (LTH).
It proposes training fixed‑mask subnetworks as Bayesian neural networks (BNNs) using Hamiltonian Monte Carlo (HMC, with NUTS) so that pruned models can reach high accuracy without the original winning‑ticket initialization.

**Strengths:**

Treating initialization as a prior and integrating over it with HMC is a coherent way to argue away initialization dependence, and the write‑up of HMC/NUTS is clear.

**Weaknesses:**

1. The paper asserts that saving the winning‑ticket initialization is undesirable but does not quantify concrete drawbacks (storage, distribution, reproducibility/operational constraints) nor provide scenarios where “mask‑only” is necessary. The case for avoiding saved initializations remains mostly qualitative.

2. HMC never runs beyond small CNNs; even ResNet‑18 cannot be loaded. No ImageNet experiments are provided.
LTH’s practical impact is greatest in large‑model/large‑data regimes.
The submission itself attributes the bottleneck to HMC’s full‑dataset requirement and heavy sampling cost.

3. The method section and Fig. 1 specify reset‑to‑initialization IMP at each pruning iteration, with no early‑checkpoint rewinding, which is a practice known to matter particularly for deeper nets.
This likely understates the lottery‑baseline on ResNet‑18/CIFAR‑10 (Table 1 shows 80.79%), making proposed methods look stronger by comparison.

4.The related‑work section mentions continuous sparsification and other efficient ticket‑finding/mask‑learning routes, but experiments omit direct comparisons to mask‑learning (e.g., continuous sparsification/L0) and dynamic sparse training (e.g., RigL/SET). Without these, the practical advantage over established initialization‑agnostic baselines is unclear.

5. SVI receives targeted hyperparameter search (KL normalization via 50‑candidate PBT), whereas comparable tuning budgets for deterministic/lottery baselines are not documented, raising fairness concerns

**Questions:**

1.Is it possible to include rewinding IMP; align training recipes with strong practice; report ≥5 seeds with 95% CIs and add significance tests. Explicitly present accuracy vs. sparsity sweeps (e.g., 90/95/98/99%) ?

2. Is it possible to provide at least one ImageNet‑scale experiment? If HMC is infeasible, demonstrate SG‑MCMC variants (e.g., SG‑HMC/SGLD) with clear trade‑offs and convergence diagnostics, since Sec. 6 suggests this as the intended path.

3. Is is possible to add SynFlow (PaI; data‑free) and RigL (DST) as primary baselines; optionally include continuous sparsification/L0?

4. Is it possible to quantify any real costs of storing initializations (storage, distribution constraints, reproducibility/IP considerations) and present scenarios where mask‑only is necessary or clearly preferable?

---

### Note · Authors · 2026-01-20

I have read and agree with the venue's withdrawal policy on behalf of myself and my co-authors.